# Volatile Flavor Improvement and Spoilage Microorganism Inhibition in Low-Salt Fish Sauce (Yulu) by Salt-Tolerant *Bacillus subtilis*

Chunsheng Li [1,2,3,*], Laihao Li [1], Shengjun Chen [1,3], Yongqiang Zhao [1,3] and Yanyan Wu [1,3]

[1] Key Laboratory of Aquatic Product Processing, Ministry of Agriculture and Rural Affairs, National R&D Center for Aquatic Product Processing, South China Sea Fisheries Research Institute, Chinese Academy of Fishery Sciences, Guangzhou 510300, China; laihaoli@163.com (L.L.); chenshengjun@scsfri.ac.cn (S.C.); zhaoyq@scsfri.ac.cn (Y.Z.); wuyygd@163.com (Y.W.)

[2] Co-Innovation Center of Jiangsu Marine Bio-industry Technology, Jiangsu Ocean University, Lianyungang 222005, China

[3] Key Laboratory of Efficient Utilization and Processing of Marine Fishery Resources of Hainan Province, Sanya Tropical Fisheries Research Institute, Sanya 572018, China

* Correspondence: lichunsheng@scsfri.ac.cn; Tel.: +86-020-89020911

**Abstract:** Use of low-salt fish sauce (Yulu) is limited due to its perishable property and rapid loss of unique flavor. In this study, a salt-tolerant strain, *Bacillus subtilis* B-2, with high protease production was used as a microbial starter for low-salt Yulu fermentation. A total of 133 volatile compounds were detected through HS-SPME-GC-MS. Most aldehydes, alcohols, ketones, furans, and hydrocarbons reached their maximum at 15 d, while most esters, aromatic compounds, acids, nitrogen compounds, and sulfur compounds peaked at 45 d. The 16S rRNA gene high-throughput sequencing showed that *Bacillus* remained in high abundance during fermentation, reaching 93.63% at 45 d. The characteristic volatile flavor was obviously improved while the microbial contamination was significantly reduced in low-salt Yulu fermented with *B. subtilis*, compared with those without a microbial starter. The correlation network map suggested that the significant decrease in *Pseudomonas*, *Achromobacter*, *Stenotrophomonas*, *Cyanobium*, *Rhodococcus*, *Brucella*, *Tetragenococcus*, and *Chloroplast* contributed most to the decreasing richness and evenness of the microbial community, while *Bacillus* was the only genus playing a key role in the inhibition of spoilage microorganisms and improvement of volatile flavor after *B. subtilis* addition. This study suggests the potential use of salt-tolerant *B. subtilis* as a special starter for industrial Yulu fermentation in the future.

**Keywords:** low-salt fish sauce; *Bacillus subtilis*; microbial community; microbial diversity; volatile flavor; correlation network map

## 1. Introduction

Fish sauce, also known as Yulu in China, is a famous aquatic condiment in Asian countries made from low-cost sea fish [1–3]. Traditional Yulu is usually fermented in high salinity (25–30%) in order to inhibit the growth of various spoilage microorganisms. However, high salt content slows down the flavor formation in Yulu because the metabolism of flavor-producing microorganisms is also suppressed in such an environment [4,5]. The fermentation cycle to obtain a good quality Yulu is rather long (1–3 years). Therefore, it is imperative to accelerate the fermentation speed of Yulu on an industrial scale.

Yulu fermentation can be accelerated using low salt. However, in a low-salt environment, Yulu's quality rapidly declines [6]. Therefore, it is necessary to add an appropriate microbial starter to suppress the growth of spoilage microorganisms. Recently, strains with high protease production, especially isolated from traditional Yulu, have been used as a microbial starter for Yulu fermentation, such as *Virgibacillus halodenitrificans* [1], *Planococcus maritimus* [7], *Halobacterium* sp. [8], and *Penicillium citrinum* [9]; these strains

can accelerate the enzymatic hydrolysis of proteins and improve the amino acid nitrogen content of Yulu. However, not many research has focused on the change in flavor of Yulu after the addition of protease-producing microbial starter though it is known that the acceleration of Yulu fermentation is often accompanied by the loss of its unique flavor. Additionally, research has also overlooked the changes in microbial community when a microbial starter is used [10,11]. Therefore, studies on these aspects are important to reveal whether a microbial starter can adapt to the Yulu fermentation environment and play a role in improving its flavor.

In our previous study, *Bacillus subtilis* B-2, a strain with high protease production, was screened from traditional Yulu [12]. This strain has high salt tolerance and stable protease production ability under high salt conditions, and has the potential to protect against the spoilage microorganisms during the fermentation of Yulu in low-salt environment. In this study, *B. subtilis* B-2 was used as the microbial starter for low-salt Yulu production. The changes in volatile compounds and microbial community during the fermentation of low-salt Yulu with this strain were analyzed using HS-SPME-GC-MS and 16S rRNA gene high-throughput sequencing, respectively. The characteristic volatile flavor compounds and microbial community in Yulu fermented with *B. subtilis* B-2 were also compared with Yulu without the microbial starter addition, followed by revealing their change mechanisms through a correlation network map. This study will be helpful in developing an effective method to improve the volatile flavor compounds and inhibit the spoilage microorganisms in low-salt Yulu fermentation.

## 2. Materials and Methods

### 2.1. Microbial Starter and Pre-Culture

The salt-tolerant *Bacillus subtilis* B-2, which was stored in the China General Microbiological Culture Collection Center (CGMCC No. 23784), was maintained on the LB agar slant (10 g/L peptone, 5 g/L yeast extract powder, 10 g/L NaCl, and 18 g/L agar; pH 7.0) at 4 °C. Pre-culture of *B. subtilis* was performed according to a previous study [13]. Briefly, the strain was first transferred on a fresh YEPD agar slant and incubated at 37 °C for 24 h. Thereafter, a loopful of the slant culture was transferred into 50 mL liquid LB medium (10 g/L peptone, 5 g/L yeast extract powder, and 10 g/L NaCl; pH 7.0) in 250 mL Erlenmeyer flask and was incubated at 37 °C and 180 r/min for 24 h.

### 2.2. Yulu Processing

After pre-culture, the strain was centrifuged for 15 min at 4 °C and 12,000× *g*. After resuspension in saline solution, the strain was used as the microbial starter to produce low-salt Yulu.

Low-salt Yulu was made from iced blue round scad (*Decapterus maruadsi*) according to the previous study [5]. Briefly, the minced whole fish was added to the fermenters with 18% NaCl and the microbial starter at $1 \times 10^6$ CFU/g fish, while the saline solution of the same volume added in the minced fish was used as the control group. Glass fermenters were used to store the minced fish and were covered with eight layers of gauze to maintain a semi-anaerobic environment. The Yulu was stirred every 5 d. After fermentation at 35 °C, the liquid sample fermented with the microbial starter was taken at 5 d (BS5), 10 d (BS10), 15 d (BS15), 30 d (BS30), and 45 d (BS45), respectively, while Yulu fermented without the microbial starter was taken at 45 d (C45) for further analysis.

### 2.3. Volatile Compound Analysis

The volatile compounds in Yulu were analyzed using HS-SPME-GC-MS [14]. Briefly, after addition of internal standard 2,4,6-trimethylpyridine (100 μL and 50 mg/L), the liquid Yulu (5.0 g) was put into the headspace vial and incubated for 10 min at 300 r/min and 60 °C. The HS-SPME was carried out using the DVB-PDMS extraction needle at 300 r/min and 60 °C for 40 min. The volatile compounds were determined on the HP-INNOWAX column (60 m × 0.25 mm × 0.25 μm) via the 7890–5977 GC-MS system (Agilent, Santa

Clara, CA, USA). The carrier gas was helium at 1.2 mL/min in the splitless mode. GC-MS was conducted as follows: 40 °C for 5 min, 40–240 °C at 5 °C /min, 240–250 °C at 10 °C /min, and 250 °C for 6 min. The ion source and quadrupole temperature was set to 230 °C and 150 °C, respectively. The scan range was a full scan of 29–450 Da. After identification in the NIST database (v17.0) using AMDIS, the concentration of each volatile compound ($C_x$) was calculated according to the following equation:

$$C_x = \frac{A_x \times M_i}{A_i \times M_Y}$$

where, $C_x$ is the concentration of each volatile compound (mg/kg), $A_x$ is the peak area of each volatile compound, $A_i$ is the peak area of 2,4,6-trimethylpyridine, $M_i$ is the weight of 2,4,6-trimethylpyridine (μg), and $M_Y$ is the weight of Yulu (g).

### 2.4. Microbial Community Analysis

The Yulu sample (20 g) was centrifuged for 10 min at 4 °C and 12,000 r/min. The microbial precipitation was used for the 16S rRNA gene high-throughput sequencing [15]. Briefly, the genomic DNA was obtained using the E.Z.N.A.® soil DNA Kit (Omega Bio-tek, Norcross, GA, USA). The primer pairs, including 338F (5′-ACTCCTACGGGAGGCAGCAG-3′) and 806R (5′-GGACTACHVGGGTWTCTAAT-3′) were used for the V3-V4 region amplification of 16S rRNA genes via PCR. Purified PCR amplicons were analyzed using paired-end sequencing on the MiSeq PE300 platform (Illumina, San Diego, CA, USA). The raw reads were demultiplexed and quality-filtered using Fastp v0.19.6, and were then merged using Flash v1.2.11. The operational taxonomic unit of 97% similarity cutoff was clustered using Uparse v7.0.1090 and was then used for taxonomy analysis via RDP Classifier v2.11.

### 2.5. Statistical Analysis

All experiments were performed in triplicate and the data were expressed as mean ± standard deviation. The heatmap of volatile compounds was developed by MetaboAnalyst v5.0 [16]. The similarity in volatile compounds among various fermentation groups was analyzed using principal component analysis (PCA). The α diversity of microbial community was determined via Mothur v1.30.2 based on the abundance of operational taxonomic unit, while PCA was used for the β diversity analysis [4]. The correlation network map was constructed using Cytoscape v.3.8.1 based on the Pearson's correlation coefficient [17].

## 3. Results and Discussion

### 3.1. Change in Volatile Compounds during Yulu Fermentation

The change in volatile compounds in Yulu during fermentation with *B. subtilis* is shown in Figure 1 and Table S1. A total of 133 volatile compounds were identified from Yulu, including 25 aldehydes, 19 esters, 17 alcohols, 17 hydrocarbons, 13 ketones, 12 aromatic compounds, 11 acids, 10 nitrogen compounds, 6 furans, and 3 sulfur compounds according to their functional groups. As shown in the PCA results (Figure 1A), three distinct groups were found during fermentation with *B. subtilis*, including early fermentation (BS5 and BS10), middle fermentation (BS15), and late fermentation (BS30 and BS45); the volatile compounds in the BS15 group were more different than the other fermentation groups. As shown in Figure 1B, the total concentration of volatile compounds in each classification increased as the fermentation went on. The total concentration of hydrocarbons, aldehydes, alcohols, ketones, and furans reached their maximum in the BS15 group, while the total concentration of esters, aromatic compounds, acids, nitrogen compounds, and sulfur compounds peaked in the BS45 group. Among the volatile compounds, hydrocarbons, aldehydes, esters, and alcohols had the highest total concentrations.

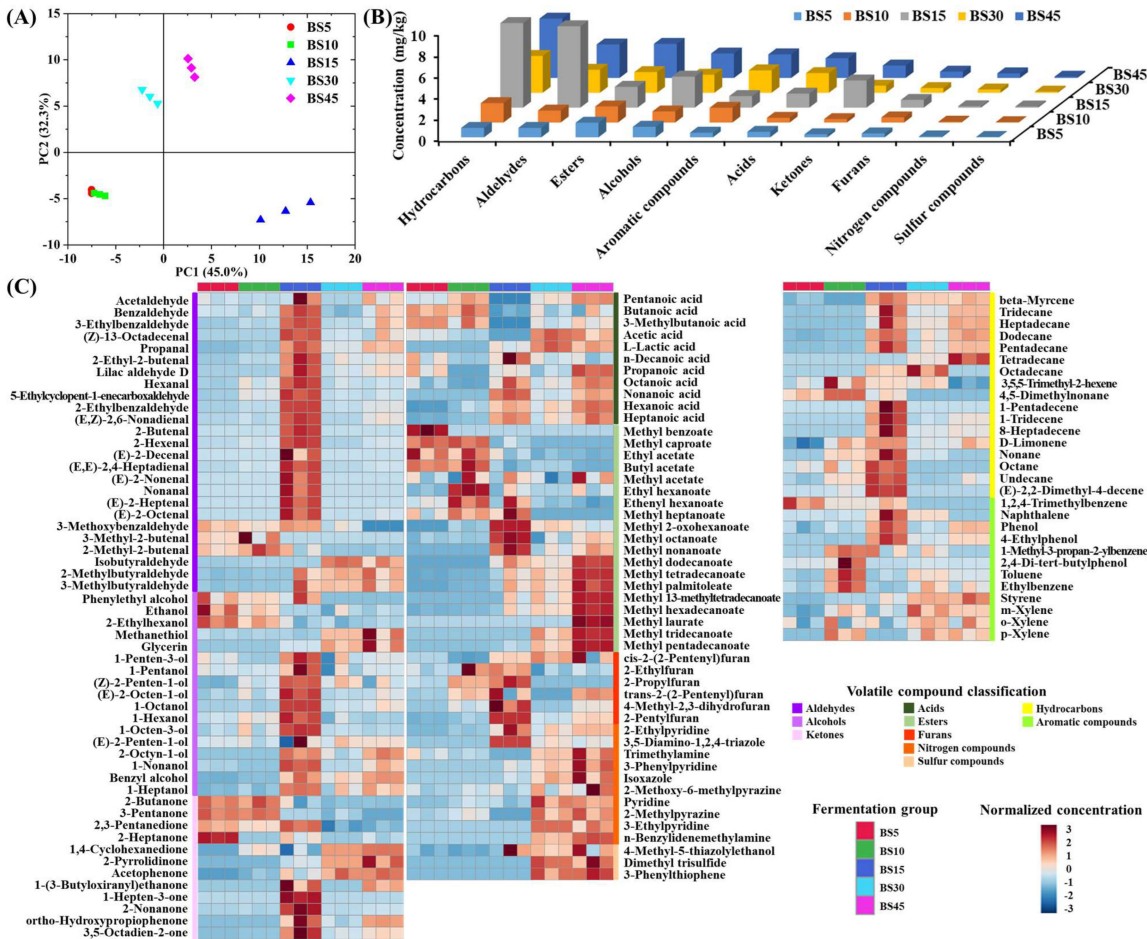

**Figure 1.** Change in volatile compounds of Yulu during fermentation with *B. subtilis*. (**A**) Similarity analysis of volatile compounds among different fermentation groups via PCA. (**B**) Change in total concentration of volatile compounds in each classification in different fermentation groups. (**C**) Heatmap of volatile compounds in each classification in different fermentation groups.

The heatmap of volatile compounds during Yulu fermentation with *B. subtilis* is presented in Figure 1C. Aldehydes, ketones, and alcohols are formed similarly through lipid oxidation or amino acid catabolism, and most of them contribute significantly to the whole flavor in various fermented foods owing to their low thresholds [4,18,19]. In this study, most aldehydes, ketones, and alcohols shared similar trends during fermentation; their highest concentrations were in middle fermentation (BS15 group), followed by the BS45 group. Various types of aldehydes were found in Yulu during fermentation with *B. subtilis*, including straight-chain aldehydes, branched aldehydes, olefine aldehydes and phenyl aldehydes. Hexanal was one of the most abundant straight-chain aldehydes, which reached 0.409 mg/kg after 15 d fermentation and remained 0.100 mg/kg at the end of fermentation. Hexanal is also reported to be abundant in Yulu [7,20]. Nonanal was another important straight-chain aldehyde which reached 1.322 mg/kg after 15 d fermentation, but disappeared after 30 d. In addition to straight-chain aldehydes, many branched aldehydes, such as 3-methylbutyraldehyde and 2-methylbutyraldehyde, peaked after 45 d fermentation, reaching 0.835 and 0.385 mg/kg, respectively. Plenty of olefine aldehydes were also found in Yulu, especially for lilac aldehyde D with the maximum concentration of 1.379 mg/kg after 15 d fermentation and 0.540 mg/kg after 45 d fermentation. The maximum concentrations of (E)-2-octenal, (E,E)-2,4-heptadienal, 2-hexenal, (Z)-13-octadecenal, and 2-ethyl-2-butenal reached 0.335, 0.239, 0.154, 0.150, and 0.116 mg/kg, respectively, in the BS15 group. Plenty of (E)-2-nonenal (0.408 mg/kg) and (E)-2-decenal (0.271 mg/kg)

were also found at 15 d, but disappeared after 30 d. Meanwhile, phenyl aldehydes were found in abundance in Yulu, especially for benzaldehyde and 3-ethylbenzaldehyde, whose maximum concentrations were 1.073 and 0.339 mg/kg in the BS15 group, followed by 0.705 and 0.174 mg/kg in the BS45 group, respectively. Benzaldehyde has also been reported to be abundant in other fermented aquatic products [21–23].

Plenty of saturated alcohols were observed in this study, especially glycerin, which increased with the increasing fermentation time and peaked in the BS45 group (0.668 mg/kg). However, glycerin is odorless, having no effect on the overall flavor of Yulu. In addition, high concentrations of 1-octanol, 1-nonanol, and 1-heptanol were observed with the maximum concentrations of 0.384, 0.307, and 0.270 mg/kg, respectively, in the BS15 group. In addition to saturated alcohols, various unsaturated alcohols were also found during Yulu fermentation, and the highest alcohols were concentrated in 1-octen-3-ol, 2-octen-1-ol, and 1-penten-3-ol with the maximum concentrations of 0.780, 0.238, and 0.205 mg/kg, respectively, in the BS15 group. Interestingly, 1-octen-3-ol and 1-penten-3-ol also widely exist in other kinds of Yulu [5,14]. In this study, 2-undecanone, 3,5-octadien-2-one, 2-nonanone, and 1,4-cyclohexanedione were abundant during fermentation of Yulu with the maximum concentrations of 1.228, 0.667, 0.302, and 0.124 mg/kg, respectively. Both 2-undecanone and 2-nonanone are also rich in the fermented tilapia sausage [15].

Acetic acid and L-lactic acid were mainly produced during carbohydrate fermentation in lactic acid bacteria [24]. In this study, high concentrations of L-lactic acid and acetic acid were found in the late fermentation of Yulu with maximum concentrations of 1.155 and 0.128 mg/kg, respectively, suggesting the high activity of lactic acid bacteria during that period. Long (C14–C18) and medium (C6–C12) chain acids are mainly formed through the hydrolysis of triglycerides and phospholipids, while hexanoic acid, octanoic acid and nonanoic acid are formed through lipid oxidation [24]. In this study, hexanoic acid and octanoic acid were mostly observed in the BS45 group, reaching 0.276 and 0.243 mg/kg, respectively, while nonanoic acid and decanoic acid peaked in the BS15 group, reaching 0.143 and 0.147 mg/kg, respectively. These acids are also abundant in the low-salt Yulu produced with the addition of *T. muriaticus*. However, most of the volatile acids have a high flavor threshold [25], causing little effect on the overall flavor of Yulu.

Volatile esters in foods are generally formed through the enzymatic condensation of acids and alcohols [26,27]. Different from aldehydes, ketones, and alcohols, most of esters in the low-salt Yulu possessed their highest concentrations in the BS45 group. Methyl tetradecanoate had the highest concentration of 1.021 mg/kg in the BS45 group. This ester is also the most abundant in the low-salt Yulu fermentation with *T. muriaticus* [5]. Although ethyl acetate reached the maximum at the beginning of fermentation (0.941 mg/kg), it remained 0.502 mg/kg at the end of fermentation. High concentrations of ethyl acetate are also found in other fermented Yulu [14,28], suggesting its key role in the characteristic flavor of Yulu. Methyl hexadecanoate, methyl palmitoleate, methyl dodecanoate, and methyl pentadecanoate also were in high abundance in Yulu, with the concentrations > 0.10 mg/kg at the end of fermentation.

A total of four furans were abundant during the fermentation with the concentrations >0.10 mg/kg at 45 d, including 2-ethylfuran, cis-2-(2-pentenyl)furan, trans-2-(2-pentenyl)furan, and 2-pentylfuran. Similarly, 2-ethylfuran and 2-pentylfuran are also abundant in other fermented aquatic products [14,29]. Most of the nitrogen compounds possessed low concentrations, except trimethylamine which reached 0.145 mg/kg at 45 d. For sulfur compounds, only the concentration of 4-methyl-5-thiazolylethanol was over 0.10 mg/kg at 45 d. Although plenty of hydrocarbons and aromatic compounds were present during fermentation with *B. subtilis*, they might contribute little to the overall characteristic flavor of Yulu owing to their high flavor thresholds [4,30]. D-Limonene was one of the highest concentrated hydrocarbons with the maximum concentration of 1.519 mg/kg in the BS15 group and 1.296 mg/kg at the end of fermentation.

### 3.2. Change in Microbial Community during Yulu Fermentation

The change in microbial community during Yulu fermentation with *B. subtilis* is shown in Figure 2. The α diversity analysis can evaluate the change in richness and evenness of microbial community [31,32]. In this study, all coverage index in the fermentation groups was over 0.999, indicating the good coverage of high-throughput sequencing (Figure 2A). As shown in Figure 2A, higher Sobs, Chao, and ACE indexes were observed in early fermentation. They all fell to the bottom in the BS30 group, suggesting minimum richness during this period. At the end of fermentation, all these three indexes slightly increased, indicating the improvement in richness of the microbial community. The Shannon index first increased and then decreased during Yulu fermentation, and it was the maximum in the BS30 group, while the Simpson index followed the opposite trend, suggesting highest evenness in the BS30 group and low evenness in the BS5 and BS45 groups (Figure 2A). The β diversity analysis can evaluate the similarity in microbial community in different samples [4]. As shown in Figure 2B, the microbial community in the B45 group had more difference than the other fermentation groups, while the microbial community in the BS5, BS10, and BS15 groups was much similar.

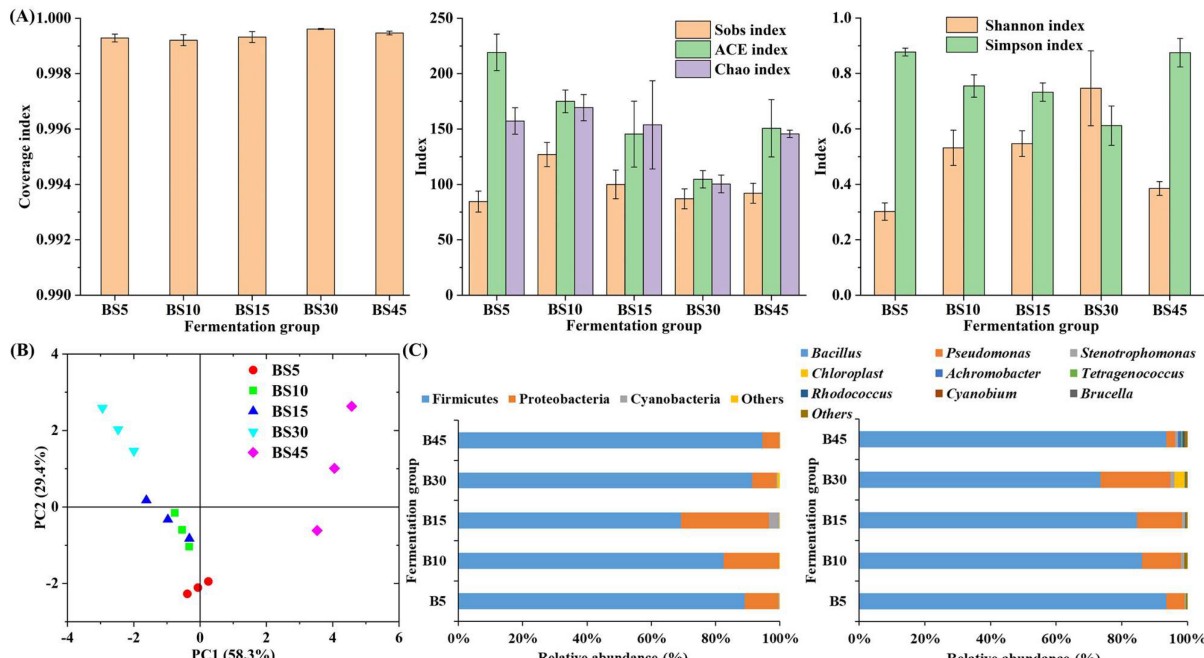

**Figure 2.** Change in the microbial community of Yulu during fermentation with *B. subtilis*. (**A**) α diversity, (**B**) β diversity, and (**C**) microbial taxonomic composition at phylum and genus levels in different fermentation groups.

The microbial composition in Yulu during fermentation with *B. subtilis* is further studied. As shown in Figure 2C, there were only three phyla with a relative abundance of over 0.1%. Firmicutes and Proteobacteria were the top phyla with a total relative abundance of over 96% in the overall fermentation. As Yulu fermentation processed, the relative abundance of Firmicutes first decreased and then increased to the maximum, reaching 94.60% at the end of fermentation, while the relative abundance of Proteobacteria followed the opposite trend. The high relative abundance of Firmicutes probably resulted from the addition of *B. subtilis*, which belongs to the Firmicutes genera. As shown in Figure 2C, there were nine microbial genera (relative abundance >0.1%) in Yulu during fermentation with *B. subtilis*. As fermentation progressed, *Bacillus* obtained the prominent position in Yulu, in that its relative abundance first decreased from 93.64% at 5 d to 73.50% at 30 d and then increased to 93.63% at 45 d. *Pseudomonas*, *Stenotrophomonas*, and *Chloroplast* showed similar trends; their relative abundance first increased and then decreased, peaking in the

B30 group, while the abundance of *Achromobacter*, *Tetragenococcus*, *Rhodococcus*, *Cyanobium*, and *Brucella* significantly increased and reached the maximum at 45 d.

### 3.3. Improvement in Volatile Flavor of Yulu after B. subtilis Addition

The characteristic volatile flavor compounds in Yulu produced with *B. subtilis* addition at the end of fermentation (BS45) were further compared with those without microbial starter addition (C45). As shown in Figure 3A, 11 volatile aldehydes were identified as the characteristic flavor aldehydes in Yulu fermented with *B. subtilis* according to the concentrations and thresholds. Interestingly, except (E)-2-octenal, benzaldehyde, and 3-ethylbenzaldehyde, all the other characteristic aldehydes were improved by *B. subtilis*, especially hexanal, 2-methylbutyraldehyde, 3-methylbutyraldehyde, lilac aldehyde D, (E,E)-2,4-heptadienal, (Z)-13-octadecenal, and 2-ethyl-2-butenal, contributing to the increase in pleasant grass, fat, and flower flavors in Yulu. A total of six volatile alcohols were identified as the characteristic flavor alcohols in the low-salt Yulu fermented with *B. subtilis*, and were all found to be improved after comparison with the C45 group. The long-chain saturated alcohols 1-octanol, 1-nonanol, and 1-heptanol all possess pleasant fat, flower, and fruit flavors with low thresholds (<0.15 mg/kg) [4]. In this study, the concentrations of these alcohols all exceeded their thresholds and were significantly enhanced by *B. subtilis*, contributing to the improvement of the overall flavor in Yulu. As an unsaturated alcohol, 1-octen-3-ol is one characteristic flavor alcohol found in plenty of fermented aquatic products [14,15,33,34] because of its low flavor threshold (0.001 mg/kg) and high concentration. In this study, the concentration of this alcohol remarkably exceeded its threshold and was significantly improved by *B. subtilis*, contributing to the improvement of mushroom flavor in Yulu. In addition, the significant increase in 1-penten-3-ol after the addition of *B. subtilis* resulted in the increase in pleasant roasted onion flavor [5] in Yulu. A total of four volatile ketones were identified as the characteristic flavor ketones in the low-salt Yulu and were all improved by *B. subtilis*. Among these ketones, 2-undecanone and 2-nonanone have the sweet, fruit, and grass flavors with low thresholds (<0.01 mg/kg) [4,15]. In this study, both ketones in the BS45 group exceeded their flavor thresholds and were obviously higher than those in the C45 group, contributing to the improvement of the overall flavor in Yulu.

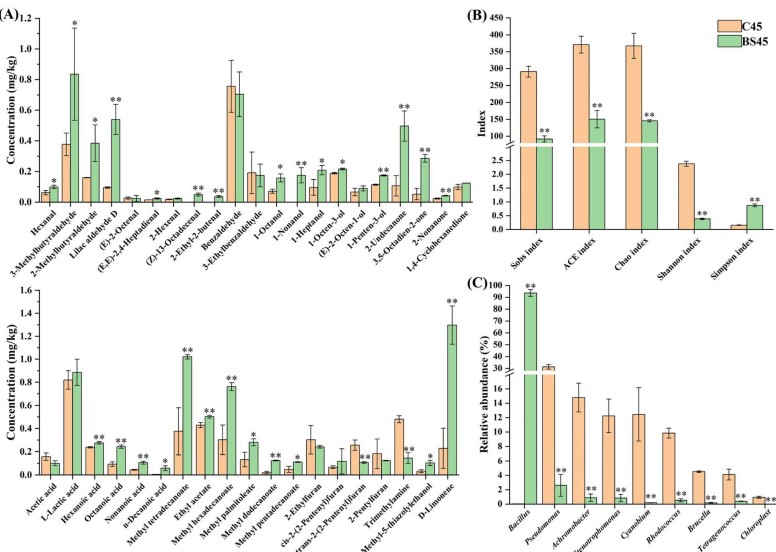

**Figure 3.** Comparison of (**A**) characteristic volatile flavor compounds, (**B**) α diversity indexes, and (**C**) microbial genera in low-salt Yulu fermented with *B. subtilis* and without microbial starter at the end of fermentation. Comparison of the difference between the C45 and BS45 groups was performed using student's *t*-test (* $p < 0.05$ and ** $p < 0.01$).

A total of six volatile acids were identified as the characteristic flavor acids in Yulu and they were all improved by *B. subtilis,* except acetic acid. Similarly, a total of six volatile esters were identified as the characteristic flavor esters in Yulu and were all improved by *B. subtilis*. The higher concentrations of methyl tetradecanoate, ethyl acetate, methyl hexadecanoate, methyl palmitoleate, methyl dodecanoate, and methyl pentadecanoate were helpful in forming the fruity and sweet flavors in Yulu. Four volatile furans were identified as the characteristic flavor furans in Yulu. Except for the significant decrease in trans-2-(2-pentenyl)furan, most of the furans showed no significant change after the addition of *B. subtilis*. Trimethylamine was the characteristic volatile flavor nitrogen compound in low-salt Yulu fermented with *B. subtilis*. Trimethylamine is produced through reduction or demethylation of trimethylamine N-oxide, especially under anaerobic conditions [35]. It is one of the main compounds in fish and fish products [5,34]. *B. subtilis* could significantly inhibit trimethylamine formation in low-salt Yulu, leading to the decrease in the unpleasant fishy flavor. Similarly, the addition of *T. muriaticus* can completely inhibit the trimethylamine formation in low-salt Yulu at the end of fermentation [5]. In low-salt Yulu fermented with *B. subtilis*, 4-methyl-5-thiazolylethanol was the only characteristic volatile flavor sulfur compound. The significant increase in 4-methyl-5-thiazolylethanol improved the fat and cooked beef flavors in Yulu. As the only one characteristic volatile flavor hydrocarbon, the significantly improved D-limonene played an important role in the improvement of orange flavor [19] in Yulu.

*3.4. Inhibition of Spoilage Microorganisms in Yulu after B. subtilis Addition*

The α diversity and microbial composition of the microbial community in low-salt Yulu fermented with *B. subtilis* at the end of fermentation (BS45) were compared with those without microbial starter addition (C45). The α diversity can effectively reveal the degree of microbial contamination in the environment [5,29]. As shown in Figure 3B, the Sobs, Chao, ACE indexes significantly decreased from 291, 371, and 367 to 92, 150, and 146 after the addition of *B. subtilis*, respectively. Similarly, the Shannon index significantly decreased from 2.38 to 0.39, while the Simpson index significantly increased from 0.16 to 0.87 after *B. subtilis* addition. These results indicated that the richness and evenness of the microbial community in Yulu were all significantly lowered by *B. subtilis* as a result of the obvious decrease in microbial contamination. Moreover, only the abundance of *Bacillus* significantly increased from 0.04% to 93.63% after the addition of *B. subtilis*, while all the other microorganisms were all inhibited, especially spoilage microorganisms, such as *Pseudomonas*, *Achromobacter*, *Stenotrophomonas*, *Cyanobium*, *Rhodococcus*, and *Brucella* with their abundance decreasing from 31.34%, 14.80%, 12.25%, 12.46%, 9.85%, and 4.51% to 2.62%, 0.91%, 0.85%, 0.19%, 0.54%, and 0.18%, respectively (Figure 3C). These results suggested that *B. subtilis* was more suitable to adapt to the Yulu fermentation environment than the other microorganisms, probably due to its salt-tolerant characteristic.

*3.5. Change Mechanism of Volatile Flavors and Spoilage Microorganisms in Yulu after B. subtilis Addition*

The correlation network map can clearly show the relational degree between different indicators [17,36,37]. In this study, the correlation network map was constructed based on the Pearson's correlation among different indexes in the BS45 and C45 groups (Figure 4). The Sobs, Chao, ACE, and Shannon indexes showed significantly negative relation with *Bacillus* but obviously positive relation with *Pseudomonas*, *Achromobacter*, *Stenotrophomonas*, *Cyanobium*, *Rhodococcus*, *Brucella*, *Tetragenococcus*, and *Chloroplast*. Meanwhile, the Simpson index exhibited the totally opposite correlation. These results indicated that the increase in *Bacillus* played a key role in decreasing microbial contamination which was mainly due to the significant decrease in *Pseudomonas*, *Achromobacter*, *Stenotrophomonas*, *Cyanobium*, *Rhodococcus*, *Brucella*, *Tetragenococcus*, and *Chloroplast* after *B. subtilis* addition. Meanwhile, except unpleasant trimethylamine, most of the characteristic pleasant volatile flavor compounds showed significantly positive correlation with *Bacillus* but were signifi-

cantly negatively related with *Pseudomonas*, *Achromobacter*, *Stenotrophomonas*, *Cyanobium*, *Rhodococcus*, *Brucella*, *Tetragenococcus*, and *Chloroplast*. It indicated that the improved volatile flavor in Yulu mainly resulted from the higher abundance of *Bacillus*. *B. subtilis* possessed good protease production ability under high salt conditions, contributing to the increase in enzymatic hydrolysis of fish proteins and production of free amino acids. As important precursors of volatile compounds, the increasing free amino acids were beneficial for the increase in volatile compounds.

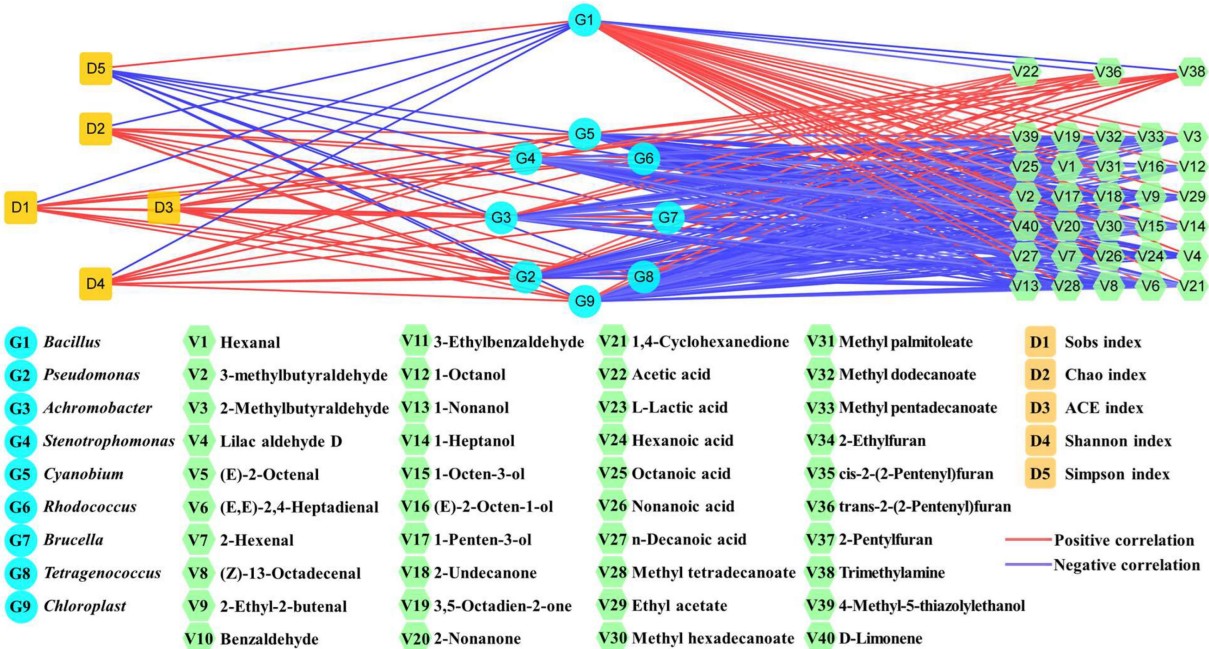

| | | | | | |
|---|---|---|---|---|---|
| G1 *Bacillus* | V1 Hexanal | V11 3-Ethylbenzaldehyde | V21 1,4-Cyclohexanedione | V31 Methyl palmitoleate | D1 Sobs index |
| G2 *Pseudomonas* | V2 3-methylbutyraldehyde | V12 1-Octanol | V22 Acetic acid | V32 Methyl dodecanoate | D2 Chao index |
| G3 *Achromobacter* | V3 2-Methylbutyraldehyde | V13 1-Nonanol | V23 L-Lactic acid | V33 Methyl pentadecanoate | D3 ACE index |
| G4 *Stenotrophomonas* | V4 Lilac aldehyde D | V14 1-Heptanol | V24 Hexanoic acid | V34 2-Ethylfuran | D4 Shannon index |
| G5 *Cyanobium* | V5 (E)-2-Octenal | V15 1-Octen-3-ol | V25 Octanoic acid | V35 cis-2-(2-Pentenyl)furan | D5 Simpson index |
| G6 *Rhodococcus* | V6 (E,E)-2,4-Heptadienal | V16 (E)-2-Octen-1-ol | V26 Nonanoic acid | V36 trans-2-(2-Pentenyl)furan | |
| G7 *Brucella* | V7 2-Hexenal | V17 1-Penten-3-ol | V27 n-Decanoic acid | V37 2-Pentylfuran | Positive correlation |
| G8 *Tetragenococcus* | V8 (Z)-13-Octadecenal | V18 2-Undecanone | V28 Methyl tetradecanoate | V38 Trimethylamine | Negative correlation |
| G9 *Chloroplast* | V9 2-Ethyl-2-butenal | V19 3,5-Octadien-2-one | V29 Ethyl acetate | V39 4-Methyl-5-thiazolylethanol | |
| | V10 Benzaldehyde | V20 2-Nonanone | V30 Methyl hexadecanoate | V40 D-Limonene | |

**Figure 4.** Correlation network map constructed using Pearson's correlation coefficient between the microbial genera and characteristic volatile flavor compounds or α diversity indexes in the BS45 and C45 groups. The red and blue lines indicated the significantly positive and negative correlation ($|r| > 0.6$ and $p < 0.05$), respectively.

## 4. Conclusions

During low-salt Yulu fermentation with *B. subtilis*, there were 133 volatile compounds, among which hydrocarbons, aldehydes, esters, and alcohols were the compounds with the highest total concentrations. Most aldehydes, alcohols, ketones, furans, and hydrocarbons reached their maximum concentrations in the BS15 group, while most esters, aromatic compounds, acids, nitrogen compounds, and sulfur compounds peaked in the BS45 group. *Bacillus* remained prominent in Yulu with a relative abundance of 93.63% at 45 d. Higher characteristic volatile flavor compounds and lower microbial contamination were found in Yulu fermented with *B. subtilis* than that in the absence of a microbial starter. The correlation network map suggested that the increase in *Bacillus* played a key role in decreasing microbial contamination which was mainly due to the significant decrease in *Pseudomonas*, *Achromobacter*, *Stenotrophomonas*, *Cyanobium*, *Rhodococcus*, *Brucella*, *Tetragenococcus*, and *Chloroplast* after *B. subtilis* addition. The improved volatile flavor in Yulu mainly resulted from the higher abundance of *Bacillus*. The salt-tolerant *B. subtilis* can perform as a potential starter to improve the volatile flavor and inhibit the spoilage microorganisms in low-salt Yulu in the future.

**Supplementary Materials:** The following supporting information can be downloaded at: https://www.mdpi.com/article/10.3390/fermentation9060515/s1, Table S1: Change of volatile compounds (mg/kg) during the fermentation of low-salt Yulu with *B. subtilis*.

**Author Contributions:** Conceptualization, C.L.; methodology, C.L.; software, C.L.; validation, S.C. and Y.Z.; formal analysis, C.L. and Y.Z.; investigation, C.L.; data curation, C.L.; writing—original draft preparation, C.L.; writing—review and editing, L.L. and Y.W.; visualization, C.L. and Y.Z.; supervision, L.L.; project administration, S.C.; funding acquisition, C.L., S.C. and Y.W. All authors have read and agreed to the published version of the manuscript.

**Funding:** This research was funded by the National Natural Science Foundation of China (32272348), the Earmarked fund for CARS (CARS-47), the Hainan Provincial Natural Science Foundation of China (322QN431), the Youth Talent Support Programme of Guangdong Provincial Association for Science and Technology (SKXRC202210), and the Central Public-interest Scientific Institution Basal Research Fund, CAFS (2020TD69 and 2020TD73).

**Institutional Review Board Statement:** Not applicable.

**Informed Consent Statement:** Not applicable.

**Data Availability Statement:** Not applicable.

**Conflicts of Interest:** The authors declare no conflict of interest.

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
