# Peer review of "Volatile Flavor Improvement and Spoilage Microorganism Inhibition in Low-Salt Fish Sauce (Yulu) by Salt-Tolerant Bacillus subtilis"

_fermentation, doi:10.3390/fermentation9060515_

Round 1

Reviewer 1 Report

The document is very well written. I have some observations.

Line 83. Put units in 5, 10, etc. (d)

Line 88. Check the spacing in "and60°C"

Author Response

Thanks for your comments. These comments are very valuable and helpful for revising and improving our paper. We have studied the comments carefully and have made correction which we hope to meet with approval.

Question 1:

Line 83. Put units in 5, 10, etc. (d)

Answer 1:

The sentence “After fermentation at 35 °C, the liquid sample was taken at 5 (BS5), 10 (BS10), 15 (BS15), 30 (BS30), and 45 (BS45) for further analysis, respectively” has been corrected to “After fermentation at 35 °C, the liquid sample fermented with microbial starter was taken at 5 d (BS5), 10 d (BS10), 15 d (BS15), 30 d (BS30), and 45 d (BS45), respectively, while the Yulu fermented without microbial starter was taken at 45 d (C45) for further analysis”.

Question 2:

Line 88. Check the spacing in "and60°C"

Answer 2:

“at 300 r/min and60 °C” has been corrected to “at 300 r/min and 60 °C”.

Reviewer 2 Report

Fermentation

Title: Volatile flavor improvement and spoilage microorganism inhibition in low-salt fish sauce (Yulu) by salt-tolerant Bacillus subtilis

Dear authors, I have some questions and I suggest some minor corrections. Good work. Thank you

1.     The description of the experimental design needs to be improved. (description of type of fish specie(s) used in this study; the description of the fermentation periods is not clear (BS5 means 5th day of fermentation? It was compared to results without microbial starter! But there is no description for the experiment without microbial starter!)

2.     The volatile compounds concentrations were obtained from an equation without using the relative response factor (kx) for each target volatile. In the analytical point of view this should be used: 

?? = kx(?? × ??)/ (?? × ??)

Authors should discuss this. 

3.     In Section 3.2. “Change of microbial community during Yulu fermentation”, some graphic or other information is missing to support the description in the text. 

4.     Across the text, authors call “classification” to the 10 different functional groups identified in the volatile organic molecules analysed. According to organic chemistry nomenclature it is “functional group” instead of “classification”.

5.     The information about GC-MS CONDITIONS is missing the indication of the split/splitless mode (split ratio ?) from ref. 14; 

Line 28, “The salt-tolerant B. subtilis is hopeful to develop as a special starter using for use in the industrial Yulu fermentation in the future.”

Line 47, What do you mean by “…improve the amino acid nitrogen of Yulu.” Explain the chemistry envolved.

Line 60, “…and high-throughput sequencing.”. This sentence is incomplete. What do you mean, sequencing of what?

Lines 75, 77, 87, 89, 98, You should always document your centrifugation procedures as g force rather than rpm.

“Section 2.2 Yulu processing”; This section should describe with more detail the low-salt fish sauce preparation. 

Line 81, The fish species used in this study should be characterized. Indicate which fish specie was used in this study, or species if more than one? Explain if it is used the whole fish or the edible parts of the fish.

Line 83, “at 5 (BS5), 10 (BS10), 15 (BS15), 30 (BS30), and 45 (BS45)” this means the interval in number of days?

Line 88, a space is missing.

Line 92, 93, the quantification equation to determine the unknown concentration should include the parameter RRF (relative response factor) of the target volatile. Why it is not considered in the quantification equation? This should be discussed.

Line 142, authors did not describe the meaning of “15 d”, “45 d”, etc. It should be described in the “Materials and Methods” section.

Line 143, a full stop is missing before “Nonanal”

Line 158, “Lots” is an unnecessary word in scientific writing (it has no meaning).

Line 221, It is not Figure 2C.

Line 223, Where is Figure 2D ? Is it Figure 2B?

Line 231, Where is Figure 2D ?

Line 231-235, This part of the discussion is not observed in the Figure 2 presented!

Line 237, Where is Figure 2E ? Is it Figure 2C?

Line 241, “…similar to the change of Firmicutes.” It is similar in the behavior but not in time. It is lower in 15 d and not in 30 d, as it is described in the text.

Line 318, Revise the spacing in the references. 

Line 358, “…age microorganisms in the low-salt Yulu in the furfure.” Future?

The english language is very good, it needs only minor corrections.

Author Response

Thanks for your comments. These comments are very valuable and helpful for revising and improving our paper. We have studied the comments carefully and have made correction which we hope to meet with approval.

Question 1:

The description of the experimental design needs to be improved. (description of type of fish specie(s) used in this study; the description of the fermentation periods is not clear (BS5 means 5th day of fermentation? It was compared to results without microbial starter! But there is no description for the experiment without microbial starter!)

Answer 1:

The description of the experimental design has been improved. The paragraph “Low-salt Yulu was made by iced blue round scad according to the previous study [5]. Briefly, the minced whole fish was added to the fermenters with 18% NaCl and microbial starter at 1×106 CFU/g fish. After fermentation at 35 °C, the liquid sample was taken at 5 (BS5), 10 (BS10), 15 (BS15), 30 (BS30), and 45 (BS45) for further analysis, respectively” has been changed to “Low-salt Yulu was made by iced blue round scad (Decapterus maruadsi) according to the previous study [5]. Briefly, the minced whole fish was added to the fermenters with 18% NaCl and microbial starter at 1×106 CFU/g fish, while the saline solution at the same volume added in the minced fish was used as the control group. Glass fermenters were used to hold the minced fish and were covered with eight layers of gauze to maintain a semi-anaerobic environment. The Yulu was stirred every 5 d. After fermentation at 35 °C, the liquid sample fermented with microbial starter was taken at 5 d (BS5), 10 d (BS10), 15 d (BS15), 30 d (BS30), and 45 d (BS45), respectively, while the Yulu fermented without microbial starter was taken at 45 d (C45) for further analysis.”.

Question 2:

The volatile compounds concentrations were obtained from an equation without using the relative response factor (kx) for each target volatile. In the analytical point of view this should be used:

?? = kx(?? × ??)/ (?? × ??)

Authors should discuss this.

Answer 2:

Thanks for your comment. This comment is very valuable. In this study, the semi-quantitative method by addition of internal standard 2,4,6-trimethylpyridine was used to determine the concentrations of volatile flavor components in Yulu. It is a mature method to compare the difference of each volatile flavor component between different samples. We will further study the absolute concentration of characteristic volatile flavor components using their standards according to the suggestion.

Question 3:

In Section 3.2. “Change of microbial community during Yulu fermentation”, some graphic or other information is missing to support the description in the text.

Answer 3:

The figure number in the text has been corrected.

“As shown in Figure 2B, higher Sobs, Chao, and ACE indexes were observed in the early fermentation” has been corrected to “As shown in Figure 2A, higher Sobs, Chao, and ACE indexes were observed in the early fermentation”.

“The Shannon index first increased and then decreased during the Yulu fermentation with the maximum in the BS30 group, while the Simpson index did the opposite, suggesting the highest evenness in the BS30 group and low evenness in the BS5 and BS45 groups (Figure 2C)” has been corrected to “The Shannon index first increased and then decreased during the Yulu fermentation with the maximum in the BS30 group, while the Simpson index did the opposite, suggesting the highest evenness in the BS30 group and low evenness in the BS5 and BS45 groups (Figure 2A)”.

“As shown in Figure 2D, different from volatile compounds” has been corrected to “As shown in Figure 2B, different from volatile compounds”.

“As shown in Figure 2D, there were only 3 phyla with the relative abundance over 0.1%” has been corrected to “As shown in Figure 2C, there were only 3 phyla with the relative abundance over 0.1%”.

“As shown in Figure 2E, there were 9 microbial genera (relative abundance >0.1%) in the Yulu during fermentation with B. subtilis” has been corrected to “As shown in Figure 2C, there were 9 microbial genera (relative abundance >0.1%) in the Yulu during fermentation with B. subtilis”.

Question 4:

Across the text, authors call “classification” to the 10 different functional groups identified in the volatile organic molecules analysed. According to organic chemistry nomenclature it is “functional group” instead of “classification”.

Answer 4:

Thanks for your suggestion. The volatile compounds were classified according to their functional groups. “according to their functional groups” has been added in the manuscript.

Question 5:

The information about GC-MS CONDITIONS is missing the indication of the split/splitless mode (split ratio ?) from ref. 14;

Answer 5:

We used the splitless mode. The GC-MS method “The carrier gas was helium at 1.2 mL/min in the splitless mode. The GC-MS was conducted as follows: 40 °C for 5 min, 40-240 °C at 5 °C /min, 240-250 °C at 10 °C /min, and 250 °C for 6 min. The ion source and quadrupole temperature was respectively set to 230 °C and 150 °C. The scan range was full scan of 29–450 Da” has been supplemented in the manuscript.

Question 6:

(1) Line 28, “The salt-tolerant B. subtilis is hopeful to develop as a special starter using for use in the industrial Yulu fermentation in the future.”

Answer: “The salt-tolerant B. subtilis is hopeful to develop as a special starter using in the industrial Yulu fermentation in the furfure” has been corrected to “The salt-tolerant B. subtilis is hopeful to develop as a special starter using for the industrial Yulu fermentation in the furfure”.

(2) Line 47, What do you mean by “…improve the amino acid nitrogen of Yulu.” Explain the chemistry envolved.

Answer: Amino acid nitrogen is the nitrogen content in the form of amino acids. Amino acid nitrogen is one of the main indicators to evaluate the quality of fish sauce.

(3) Line 60, “…and high-throughput sequencing.”. This sentence is incomplete. What do you mean, sequencing of what?

Answer: “high-throughput sequencing” has been changed to “16S rRNA gene high-throughput sequencing”.

(4) Lines 75, 77, 87, 89, 98, You should always document your centrifugation procedures as g force rather than rpm.

Answer: 180 r/min and 300 r/min are the speed of shaker or oscillator rather than centrifuge. “12000 r/min” has been changed to “12000 × g”.

(5) “Section 2.2 Yulu processing”; This section should describe with more detail the low-salt fish sauce preparation.

Answer: The fish sauce preparation with more detail “Low-salt Yulu was made by iced blue round scad (Decapterus maruadsi) according to the previous study [5]. Briefly, the minced whole fish was added to the fermenters with 18% NaCl and microbial starter at 1×106 CFU/g fish, while the saline solution at the same volume added in the minced fillet was used as the control group. Glass fermenters were used to hold the minced fish and were covered with eight layers of gauze to maintain a semi-anaerobic environment. The Yulu was stirred every 5 d.” have been added in the manuscript.

(6) Line 81, The fish species used in this study should be characterized. Indicate which fish specie was used in this study, or species if more than one? Explain if it is used the whole fish or the edible parts of the fish.

Answer: We used iced blue round scad (Decapterus maruadsi) and the whole fish which have been stated in the manuscript.

(7) Line 83, “at 5 (BS5), 10 (BS10), 15 (BS15), 30 (BS30), and 45 (BS45)” this means the interval in number of days?

Answer: “at 5 (BS5), 10 (BS10), 15 (BS15), 30 (BS30), and 45 (BS45)” has been corrected to “at 5 d (BS5), 10 d (BS10), 15 d (BS15), 30 d (BS30), and 45 d (BS45)”.

(8) Line 88, a space is missing.

Answer: “at 300 r/min and60 °C” has been corrected to “at 300 r/min and 60 °C”.

(9) Line 92, 93, the quantification equation to determine the unknown concentration should include the parameter RRF (relative response factor) of the target volatile. Why it is not considered in the quantification equation? This should be discussed.

Answer: Thanks for your comment. In this study, the semi-quantitative method by addition of internal standard 2,4,6-trimethylpyridine was used to determine the concentrations of volatile flavor components in Yulu. It is a mature method to compare the difference of each volatile flavor component between different samples. We will further study the absolute concentration of characteristic volatile flavor components using their standards according to the suggestion.

(10) Line 142, authors did not describe the meaning of “15 d”, “45 d”, etc. It should be described in the “Materials and Methods” section.

Answer: “After fermentation at 35 °C, the liquid sample fermented with microbial starter was taken at 5 d (BS5), 10 d (BS10), 15 d (BS15), 30 d (BS30), and 45 d (BS45), respectively, while the Yulu fermented without microbial starter was taken at 45 d (C45) for further analysis” has been described in the “Materials and Methods” section.

(11) Line 143, a full stop is missing before “Nonanal”

Answer: a full stop has been added before “Nonanal”.

(12) Line 158, “Lots” is an unnecessary word in scientific writing (it has no meaning).

Answer: “Lots of” has been corrected to “Plenty of”.

(13) Line 221, It is not Figure 2C.

Answer: “Figure 2C” has been corrected to “Figure 2A”.

(14) Line 223, Where is Figure 2D ? Is it Figure 2B?

Answer: “Figure 2D” has been corrected to “Figure 2B”.

(15) Line 231, Where is Figure 2D ?

Answer: “Figure 2D” has been corrected to “Figure 2C”.

(16) Line 231-235, This part of the discussion is not observed in the Figure 2 presented!

Answer: This part of the discussion is shown in the Figure 2C.

(17) Line 237, Where is Figure 2E ? Is it Figure 2C?

Answer: “Figure 2E” has been corrected to “Figure 2C”.

(18) Line 241, “…similar to the change of Firmicutes.” It is similar in the behavior but not in time. It is lower in 15 d and not in 30 d, as it is described in the text.

Answer: “similar to the change of Firmicutes” has been deleted.

(19) Line 318, Revise the spacing in the references.

Answer: The spacing in the references was shown according to the journal requirement.

(20) Line 358, “…age microorganisms in the low-salt Yulu in the furfure.” Future?

Answer: “in the furfure” has been corrected to “in the future”.

Reviewer 3 Report

The manuscript received for review investigates the possibility to develop an effective method to enchance the volatile flavor compounds and preserve low-salt Yulu by inhibiting spoilage microorganisms’ development, by implementing salt-tolerant Bacillus subtilis, as a starter culture.

Manuscript is clearly and adequately written.

The title of the manuscript is informative and provides sufficient data.

Introduction section comprehensive, with enough contemporary literature.

The Materials and Methods section describes samples’ preparation process and all conducted analysis in sufficient details.

The results and discussion section is appropriate, detail with adequate referencing to other authors’ findings. This section presents the changes of volatile compounds and microbial community during Yulu fermentation, improvement of volatile flavor, inhibition of spoilage microorganisms and change mechanism of volatile flavor and spoilage microorganisms in the Yulu by B. subtilis.

All conducted analysis provide comprehensive insight in changes that occur, and applied statistical methods discovers and depicts underlying mechanisms.  

Conclusion section sums the presented results in appropriate manner.

Some minor corrections are noted in the pdf file.

Reviewer recommendation: Minor revision.

Author Response

Thanks for your comments. These comments are very valuable and helpful for revising and improving our paper. We have studied the comments carefully and have made correction which we hope to meet with approval.

Question 1:

Be more specific, provide units for: 5, 10, 15, 30, 45.

Answer 1:

“After fermentation at 35 °C, the liquid sample was taken at 5 (BS5), 10 (BS10), 15 (BS15), 30 (BS30), and 45 (BS45) for further analysis, respectively.” has been corrected to “After fermentation at 35 °C, the liquid sample fermented with microbial starter was taken at 5 d (BS5), 10 d (BS10), 15 d (BS15), 30 d (BS30), and 45 d (BS45), respectively, while the Yulu fermented without microbial starter was taken at 45 d (C45) for further analysis”.

Question 2:

This statement is more suitable for conclusion section. Rephrase, or move to conclusion.

Answer 2:

The sentence “In this study, the salt-tolerant B. subtilis that exhibited good ability to inhibit spoilage microorganisms and improve volatile flavor, was hopeful to develop as a special starter for the industrial Yulu fermentation in the future” has been moved to conclusion according to the suggestion.